# Enzymatic Hydrolysate of Cinnamon Waste Material as Feedstock for the Microbial Production of Carotenoids

**DOI:** 10.3390/ijerph18031146

**Published:** 2021-01-28

**Authors:** Stefano Bertacchi, Stefania Pagliari, Chiara Cantù, Ilaria Bruni, Massimo Labra, Paola Branduardi

**Affiliations:** 1BioIndTechLab, Department of Biotechnology and Biosciences, University of Milano—Bicocca, 20126 Milan, Italy; s.bertacchi@campus.unimib.it (S.B.); c.cantu@campus.unimib.it (C.C.); 2ZooPlantLab, Department of Biotechnology and Biosciences, University of Milano—Bicocca, 20126 Milan, Italy; stefania.pagliari@unimib.it (S.P.); ilaria.bruni@unimib.it (I.B.); massimo.labra@unimib.it (M.L.)

**Keywords:** microbial-based bioprocesses, cinnamon waste, separate hydrolysis and fermentation (SHF), *Rhodosporidium toruloides*, carotenoids

## Abstract

In the context of the global need to move towards circular economies, microbial cell factories can be employed thanks to their ability to use side-stream biomasses from the agro-industrial sector to obtain additional products. The valorization of residues allows for better and complete use of natural resources and, at the same time, for the avoidance of waste management to address our needs. In this work, we focused our attention on the microbial valorization of cinnamon waste material after polyphenol extraction (C-PEW) (*Cinnamomum verum* J.Presl), generally discarded without any additional processing. The sugars embedded in C-PEW were released by enzymatic hydrolysis, more compatible than acid hydrolysis with the subsequent microbial cultivation. We demonstrated that the yeast *Rhodosporidium toruloides* was able to grow and produce up to 2.00 (±0.23) mg/L of carotenoids in the resulting hydrolysate as a sole carbon and nitrogen source despite the presence of antimicrobial compounds typical of cinnamon. To further extend the potential of our finding, we tested other fungal cell factories for growth on the same media. Overall, these results are opening the possibility to develop separate hydrolysis and fermentation (SHF) bioprocesses based on C-PEW and microbial biotransformation to obtain high-value molecules.

## 1. Introduction

Biobased processes involve the exploitation of different renewable biomasses that possess a faster turnover compared to fossil resources and therefore have a reduced impact on the environment. The forestry, agriculture, and food industries are the main sectors involved in bioeconomy worldwide [1]. Sustainability and circularity are key features in this scenario, with the obtainment of products and energy from biomasses: indeed, the main aim of biorefineries is to exploit such resources as alternatives to fossil ones [2]. Agricultural biomasses are generally related to staple crops (e.g., cereals, tubers, corn, and sugarcane), which can satisfy the issue of circularity but cannot match all the criteria of sustainability. Therefore, there is an increasing need for utilizing side-stream materials that can be valorized in second-generation biorefineries, often by means of the so-called microbial cell factories [3,4]. These side-streams materials can be derived not only from staple crop processing but also from minor crops (for example, spice crops), which are often directly used in the food industry.

One of the main interesting strategies to ennoble minor crops consists in the extraction of bioactive molecules by using chemical or biotechnological processes with low environmental impact [5,6]. In the case of the presence of residual biomasses after processing, these are mostly used to produce energy, which is a low-added value product usually obtained starting from abundant biomasses to compensate costs and market requirements.

Embedded in this approach, there is the aim to transform the entire biomass into a resource, potentially exploiting all the different portions of the plant: depending on composition and volumes, different innovative processes can be developed to obtain high added-value compounds, especially from these minor crops. 

Given these considerations, in this work, we focused our attention on cinnamon (*Cinnamomum verum* J.Presl sin. *C. zeylanicum* Blume), a plant of Asian origin that has been exploited as a spice for centuries, with many nutraceutical properties and largely adopted as a food ingredient. At the botanical level, cinnamon bark is widely used by industries for the extraction of essential oils and secondary metabolites such as polyphenols to be used in food preparation, fragrances, and perfumes [7]. In these sectors, cinnamon can be deployed as an additive for both exploiting its antimicrobial properties and increasing the nutritional values of foods [8,9,10]. The global demand for cinnamon has been increasing during the last years also due to its possible antioxidant, anti-inflammatory, and antitumoral applications [11]. As a result of industrial processing, large quantities of waste vegetable matrix rich in cellulose and lignin [12] are obtained annually. Nevertheless, this loss has the potential to become a new resource as feedstock for the development of bioprocesses based on so-called microbial cell factories.

Indeed, microorganisms can transform different raw materials of plant (or animal) origin into valuable molecules such as biofuels, chemical platforms, biopolymers, and nutraceuticals [13]. Because of its lignocellulosic nature, the cellulose and hemicellulose content of cinnamon is an unmissable opportunity to be exploited as the main carbon and energy source for microbial growth possibly related to the biosynthesis of interest, given that other residual bioactive compounds are not expressing their antimicrobial properties. In fact, because of the presence of several antimicrobial molecules (i.e., cinnamaldehyde and cinnamic acid [14,15]), cinnamon-derived biomasses have never been considered as feedstock to be deployed in bioprocesses based on microbial cell factories; therefore, their biotechnological potential still needs to be explored.

For these reasons, the aim of this study focuses on the development of a novel bioprocess (*i*) by establishing and optimizing enzymatic hydrolysis of the cinnamon bark (CB) and of the derived residual biomass (polyphenols extraction waste, C-PEW) with varying chemicophysical parameters; (*ii*) by evaluating the criticisms in bioprocesses such as inhibitory molecules for microbial and, in particular, antifungal growth; and (*iii*) by identifying a potential cell factory and a product of interest. 

With respect to the process developed, the oleaginous yeast *Rhodosporidium toruloides* was identified as a cell factory. It is naturally able to produce and accumulate carotenoids, valuable compounds with vast markets, with the food and feed sectors being the prominent ones [16,17,18]. In the present work, the enzymatic hydrolysis of C-PEW was matched with the ability of *R. toruloides* to produce carotenoids, and a separate hydrolysis and fermentation (SHF) process was run. The obtainment of 2.0 ± 0.23 mg/L of carotenoids starting from 9% (*w/v*) of initial biomass with a yield of 0.0053 ± 0.0006% on total sugars provided is a successful proof of concept of the possibility to provide an additional added value to the original biomass beyond the conventional extraction of nutraceuticals from cinnamon. Lastly, considering the promising results obtained on carotenoids production, other potential cell factories were tested for their ability to grow on C-PEW hydrolysate, expanding the industrial implications of our findings.

## 2. Materials and Methods

### 2.1. Plant Biomass: Feedstock Preparation and Composition

Epo S.r.l., Milano, Italy, provided cinnamon bark (*Cinnamomum verum* J.Presl, sin. *C. zeylanicum* Blume) grown in Madagascar. The cinnamon bark (CB) was stored at 25 °C away from heat and light sources until its use. Before extraction, the cinnamon bark was pulverized with an electric laboratory grinder. The cinnamon bark underwent an extraction process with water at 60 °C as described by [19] to obtain polyphenols. This extraction protocol was comparable with the one used by Epo S.r.l. in order to simulate their industrial process and to obtain realistic waste material at the laboratory scale. At the end of the process, the cinnamon waste material after polyphenol extraction (C-PEW) was recovered, dried out to eliminate the water derived from the extraction, and stored at −20 °C until its use. To measure the water percentage of CB, 0.9 g of biomass was dried out at 160 °C for 3 h and then weighed again to calculate the amount of evaporated water. The biomass was heat-incubated for additional 3 h to assess a lack of further changes in weight compared to the initial treatment. To analyze the chemical composition of CB and C-PEW, the biomass was processed following the protocol for the analysis of structural carbohydrates and lignin in the biomass of the National Renewable Energy Laboratory (NREL, https://www.nrel.gov/docs/gen/fy13/42618.pdf) with modifications as follows: 300 mg of biomass were diluted in 3 mL H_2_SO_4_ 72% (*v/v*), and then incubated at 30 °C for 1 h, stirring thoroughly every 10 min. The solution was diluted to 4% (*v/v*) by adding 84 mL of distilled water, mixed by inversion and autoclaved (121 °C, 1 h). The hydrolysis mixture was vacuum filtered through one of the previously weighted filtering crucibles, and the insoluble components were measured gravimetrically on the filter paper. The filtered liquid was neutralized with NaOH to pH 5–6, and then, the samples were analyzed by High-Performance Liquid Chromatography (HPLC) (as described below). Three independent experimental replicates were performed.

### 2.2. Pretreatment and Enzymatic Hydrolysis of Cinnamon Bark and Waste Material

Enzymatic hydrolysis of the cinnamon bark powder and waste material was performed using the enzyme cocktail NS22119, kindly provided by Novozymes (Novozymes A/S, Copenhagen, Denmark). As described by the manufacturer, NS22119 contains a wide range of carbohydrases, including arabinase, β-glucanase, cellulase, hemicellulase, pectinase, and xylanase from *Aspergillus aculeatus*; 9% (*w/v*) of cinnamon bark and waste material were steeped in water with a final volume of 30 mL and then autoclaved (121 °C, 1 h) in order to both sterilize and pretreat the biomass. Although mild physical pretreatment by autoclaving is less effective than chemical pretreatments, we decided not to involve their use because the overall processing, starting from the phenolic extraction that occurs upstream (and generate this waste), is intended to minimize environmental impacts.

Afterwards, the enzymes (11.9% *w/wbiomass*) were added directly in the solution and incubated at pH 5.5 and at 50 °C in a water bath under agitation (105 rpm). One milliliter of the sample was collected every 2, 4, and 6 h from the start, and the sugar content was analyzed by HPLC (see below). A high dosage provides an indication of the maximum enzymatically accessible sugar content, although low enzyme dosages provide a target for commercially feasible hydrolysis for further developments. Three independent experiments were performed.

### 2.3. Microbial Strains and Media

*S. cerevisiae* CEN.PK 102-5B was obtained from Peter Kötter (Institut fur Mikrobiologie der Johann Wolfgang Goethe Universitat, Frankfurt, Germany). The other strains used were *Komagataella phaffii* X-33 (formerly *Pichia pastoris,* ThermoFisher Scientific, Waltham, MA, USA); *Scheffersomyces stipitis CBS 6054* (CBS Fungal Biodiversity Centre, Utrecht, The Netherlands); *Zygosaccharomyces bailii* ATCC 8766 and *Zygosaccharomyces parabailii* ATCC 60483 (ATCC Virginia, Manassas, VA, USA); *Rhodosporidium toruloides* DSM 4444, *Rhodotorula glutinis* DSM 10134, *Cryptococcus curvatus* DSM 70022, and *Aureobasidium pullulans* DSM P268 from DSMZ (German Collection of Microorganisms and Cell Cultures, GmbH, Braunschweig, Germany); and *Kluyveromyces marxianus* NBRC1777 (Biological Resource Center, NITE, NBRC, Tokyo, Japan). All the strains were stored in cryotubes at −80 °C in 20% glycerol (*v/v*) and were pre-inoculated at 30 °C on rich medium plates: 1% yeast extract (Biolife Italia S.r.l., Milan, Italy), 2% peptone, and 2% glucose (Sigma-Aldrich Co., St Louis, MO, USA). The enzymatic hydrolysate from C-PEW was used for formulating the media for the plates in order to test the ability of the aforementioned strains to grow in such substrates. This feature was qualitatively assessed after 72 h of growth at 30 °C.

### 2.4. R. Toruloides Cultivation and Carotenoids Production

*R. toruloides* was pre-inoculate in a synthetic medium constituting (per liter) 1 g of the yeast extract, 1.31 g of (NH_4_)_2_SO_4_, 0.95 g of Na_2_HPO_4_, 2.7 g of KH_2_PO_4_, and 0.2 g of Mg_2_SO_4_·7H_2_O and was supplemented with 15 g/L of glycerol as the main carbon source and a 100× trace mineral stock solution as follows (per liter): 4 g of CaCl_2_·2H_2_O, 0.55 g of FeSO_4_·7H_2_O, 0.52 g of citric acid, 0.10 g of ZnSO_4_·7H_2_O, 0.076 g of MnSO_4_·H_2_O, and 100 μL of 18 M H_2_SO_4_. The yeast extract was purchased from Biolife Italia S.r.l., Milan, Italy. All other reagents were purchased from Sigma-Aldrich Co., St Louis, MO, USA. Pre-inoculum was run in that medium until stationary phase; then, cells were inoculated at an optical density (OD, 600 nm) of 0.2 in shake flasks at 30 °C and 160 rpm with C-PEW hydrolysate. After enzymatic hydrolysis, C-PEW hydrolysate was centrifuged at 4000 rpm for 10 min to separate the water insoluble components: the liquid medium obtained was used for the growth and the production of carotenoids by *R. toruloides*. Three independent experiments were performed.

### 2.5. Analytical Methods

HPLC analyses were performed to quantify the amount of glucose, sucrose, arabinose, fructose, and acetic acid. One milliliter of the liquid fraction was collected from each enzymatic hydrolysis, centrifuged twice (7000 rpm, 7 min, and 4 °C), and then analyzed at the HPLC using a Rezex ROA-Organic Acid (Phenomenex). The eluent was 0.01 M H_2_SO_4_ pumped at 0.5 mL min^−1^, and column temperature was 35 °C. Separated components were detected by a refractive-index detector, and peaks were identified by comparison with known standards (Sigma-Aldrich, St Louis, MO, USA). HPLC analyses were performed to quantify also the amount of cinnamic acid, cinnamaldehyde, 4-hydroxybenzoic acid, and p-coumaric acid using the analysis protocol reported by [20]. The extracts were previously filtered with a 0.22 μm polytetrafluoroethylene (PTFE) filter. The column used for this chromatographic stroke was an Agilent Zorbax SB-C18 (4.6 × 250 mm, 5 μm) composed of carbon chains of 18 carbon atoms and set to a temperature of 30 °C. Two mobile phases were used: phase A (aqueous solution of phosphoric acid H_3_PO_4_ at pH 3) and phase B (acetonitrile CH_3_CN, 99.98% pure, HPLC grade). The volume of samples injected for analysis was 50 μL. The biomolecules were monitored by setting the diode array detector (DAD) detection signal to 280 nm.

The cellular dry weight (CDW) was measured gravimetrically after drying 1 mL of cell culture (Concentrator 5301, Eppendorf AG, Hamburg, Germany). The titer of carotenoids extracted in acetone from *R. toruloides*, as reported in [18], was determined spectrophotometrically (UV1800; Shimadzu, Kyoto, Japan) based on the maximum absorption peak for β-carotene (455 nm). A calibration curve with standard concentration of β-carotene was used for quantification.

A Urea/Ammonia Assay Kit (K-URAMR, Megazyme International Limited, Bray, Ireland) was used to determine the amount of ammonia and urea in the C-PEW hydrolysate.

The pH was measured with indicator strips in order to assess if the conditions were suitable for the enzymatic hydrolysis and to foresee possible detrimental effects on microbial growth of the final media.

### 2.6. Calculations

Sugar recovery (S_r_) was calculated as a percentage of the sugar yield by enzymatic hydrolysis (Y_EH_) when compared with the yield obtained from the total acid hydrolysis of the biomass (Y_AH_).
Sr=YEHYAH×100

For statistical analysis, heteroscedastic two-tailed *t* test was applied.

## 3. Results and Discussion

### 3.1. Evaluation of Total Composition of the CB and C-PEW by Acid Hydrolysis

The provided cinnamon bark (CB) and the related waste material derived from polyphenol extraction (C-PEW) were subjected to total acid hydrolysis to assess their composition in terms of water, insoluble components, acetate, and sugars. Insoluble components and acetate are among the main growth inhibitors commonly linked to residual biomasses, whereas sugars act as the main carbon and energy sources for microbial cell factories. As shown in Table 1, CB and C-PEW showed similar compositions among all the analyzed constituents, consistent with the fact that the components previously extracted (i.e., polyphenols) constitute only a limited amount of the whole biomass. These data are also in accordance with the few previously reported examples from both *C. verum* (formerly *C. zeylanicum*) and *C. cassia* [11,21]. Remarkably, the vast majority of sugars is composed by glucose (more than 65% *w/w*), being up to 27% *w/w* of the total biomass. 

Total acid hydrolysis was then no longer used for the experiments as it creates conditions that are detrimental (if not incompatible) with microbial growth and requires processing conditions that have a higher environmental impact if compared with enzymatic hydrolysis. In addition, its use is limited by the release of inhibitory compounds and a very low final pH that would need a neutralization step prior to use as the growth media [22,23].

### 3.2. Enzymatic Hydrolysis of CB and C-PEW and Composition of the Hydrolysates

After assessing the composition of both CB and C-PEW, enzymatic hydrolysis of such biomasses was performed. To the best of our knowledge, there is only a single accessible peer-reviewed example of enzymatic hydrolysis of cinnamon or cinnamon-derived biomasses [21], and it is not related to the development of a microbial-based bioprocess. We first assessed the effect of enzymatic cocktail NS22119 11.9% *w/wbiomass* on CB or C-PEW 9% *w/v*: the exceeding amount of enzyme was proposed to maximize the hydrolysis of this novel substrate. The pH of both CB and C-PEW were measured between the pre-treatment and the hydrolysis step obtaining pH of 4.5 and 3.5, respectively. Both these values are lower than the optimum of the enzymatic cocktail (pH 6, as described by the manufacturer); therefore, we tested the saccharification at both the initial and the optimal pH. 

As shown in Figure 1A, the enzymatic cocktail hydrolyzed CB at both pH during the first hours of saccharification. Prolonging the incubation beyond 6 h up to 24 h did not significantly increase the sugar release (data not shown): these observations are in accordance with previous applications of this enzymatic cocktail on a different residual biomass [18]. As shown in Table 2, in both cases, glucose is the main sugar released, in accordance with the data obtained from the total acid hydrolysis (Table 1). Figure 1A shows that the release of sugars from CB is higher (13%) when performed at pH 6 (*p* < 0.05). Despite a technoeconomic analysis not yet performed, it is reasonable to conclude that this increase would not justify the additional neutralization step, considering that the difference was constituted by about 1 g/L of total sugars. 

As shown in Figure 1B, increasing the pH did not affect the enzymatic hydrolysis of C-PEW. In addition, the sugar released from CB and C-PEW, at their original pH, are comparable: this means that polyphenol extraction did not impair the enzymatic activity. In order to reduce the use of neutralizing agents and therefore the operative steps, increasing both the economic and environmental sustainability of the process, the hydrolysis of C-PEW at pH 3.5 was chosen as the preferred one. In these conditions, considering the original sugar content, the yield of sugars from C-PEW was 33% on the total available, with a yield of glucose of 36.7% on the total available (see the Calculation subsection in the Materials and Methods section). In addition, 8.1 ± 2.1 mg/L of ammonia and 4.5 ± 1.35 mg/L of urea, which may act as nitrogen sources, were detected in the C-PEW hydrolysate. 

The presence of aromatic compounds typical of cinnamon was evaluated too because of their known antimicrobial activity, concentrated in their essential oils. For example, cinnamaldehyde is known to possess a fungicidal activity [11,24,25]. Therefore, we investigated its presence together with cinnamic acid, p-coumaric acid, and 4-hydroxybenzoic acid in the hydrolysate without adjusting pH, since these molecules may impair microbial growth as well [22], reducing in return the applications of biomasses derived from the cinnamon supply chain. As shown in Table 3, cinnamaldehyde and cinnamic acid were the main aromatic compounds detected in the hydrolysate of CB. When compared with the same biomass hydrolyzed without the pretreatment, a reduction in the titer of most of these molecules can be observed: heat is indeed known to have this kind of effect on such volatile compounds [11,26,27]. The titer of cinnamic acid and cinnamaldehyde detected in the C-PEW hydrolysate was inferior compared to the CB one (Table 3): this observation is consistent with the polyphenol extraction that cinnamon was subjected to. 

Taken together, these data confirmed that the C-PEW hydrolysate was comparable (in terms of sugars released) with the hydrolysate of the original edible biomass (CB), with an important reduced amount of growth inhibitors. Therefore, we excluded CB from the following experiments. Indeed, since C-PEW is a residual biomass not further valorized, it is a suitable feedstock to be considered in a microbial-based bioprocess: its hydrolysis can provide glucose as the main accessible carbon source, even without pH corrections.

### 3.3. Production of Carotenoids from C-PEW Hydrolysate

In order to valorize C-PEW as feedstock biomass in a bioprocess, we focused our attention on the yeast *Rhodosporidium toruloides*, known for being able to withstand residual biomasses when used as feedstock for both their growth and the production of carotenoids [16,18]. When tested on plates prepared with C-PEW as the growth medium, *R. toruloides* was able to grow and accumulate pigments (see Section 3.4); therefore, we tested the production in shake flasks as well. The type of process chosen was a separate hydrolysis and fermentation (SHF) in order to eliminate the water insoluble components (WIS) from the media. Figure 2 shows the growth of *R. toruloides* on sugars released from C-PEW hydrolysis at pH 3.5 and the co-current production of carotenoids. In particular, we could observe that the production reached its maximum after 48 h in concomitance with the beginning of the stationary phase. These data are in accordance with previous reports that carotenogenic microorganisms like *R. toruloides* produce carotenoids mainly in response to stressful or suboptimal conditions such as the stationary phase itself. In addition, only a limited amount of sugars was consumed: to analyze this phenomenon, we provided to *R. toruloides* the synthetic medium supplemented with the same amount of sugars present in C-PEW hydrolysate, setting the pH value at 3.5 or 5.5 (optimal for this yeast). As shown in Appendix A, *R. toruloides* was able to completely consume the sugars and to reach a higher CDW than in C-PEW hydrolysate regardless of the initial pH. The reduced growth and sugar consumption observed in the C-PEW hydrolysate could be ascribed to the low initial amount of nitrogen source, which resulted in depletion after 48 h (Appendix A), in correspondence to the stationary phase of growth. As the inhibitory compounds typical of cinnamon origin were not detected after 24 h, nitrogen depletion should be considered the main bottleneck for *R. toruloides* growth in C-PEW hydrolysate.

In SHF conditions with 9% (*w/v*) C-PEW hydrolysate, *R. toruloides* was able to accumulate up to 2.0 ± 0.23 mg/L of carotenoids after 48 h of growth, with a productivity of 0.04 ± 0.005 g/L/h and yields of 0.09 ± 0.007% on CDW, of 0.11 ± 0.013% on consumed sugars, and of 0.005 ± 0.0006% on the total sugars provided. When considering high added-value products, yields are not crucial but important to describe the process itself and to understand how to further improve it. In fact, although this titer of carotenoids is comparable with other recently reported bioprocesses based on the combination of *R. toruloides* and residual biomasses, like *Camelina sativa* meal, carob pulp syrup, sugarcane bagasse, and molasses [18,28,29], the possibility to increase glucose consumption might ameliorate the result obtained here.

This work focused on the valorization of cinnamon by-products by their exploitation as a substrate for microorganisms. Industries in this sector must eliminate large amounts of residues each year from the processing of cinnamon extraction of polyphenols and other residual biomasses derived from biomolecule extraction. Although this waste disposal cost may not be excessively high, it is important to consider it in the scenario of the reconversion to a circular economic model fostered by the European Union, promoting increasing attention towards the exploitation of industrial products following a biorefinery model, which aims to obtain more products starting from a single biomass [30,31,32]. This is also in line with the ambitious aims and goals of the Green Deal [33], where every activity can contribute to reaching the declared “carbon neutral” status. Additionally, other microbial cell factories might be deployed in bioprocesses involved in the conversion of C-PEW to compounds of industrial relevance, such as, as a matter of example, lipids and shikimate from species such as *Cryptococcus curvatus* and *Scheffersomyces stipitis*. 

### 3.4. Testing the Growth of Other Fungal Cell Factories on C-PEW Hydrolysate 

In order to widen the possible outcomes of the exploitation of C-PEW, we tested the ability of other yeast cell factories to grow on such biomass. Testing of possible candidates was needed since, to the best of our knowledge, there were no previous examples of cinnamon use for such purposes. In addition, cinnamon is known to possess a potent antimicrobial activity [11]; therefore, putative cell factories must be tested and selected for their ability to grow on this new medium. In fact, cinnamon antimicrobial activity has been studied with regard to several pathogenic microorganisms: in particular, the minimal inhibitory concentrations of cinnamaldehyde were 125 ± 25 g/L for *Candida albicans*, 88 ± 13 g/L for *Aspergillus niger*, 300 ± 0 g/L for *Escherichia coli*, and 250 ± 50 g/L for *Staphylococcus aureus* [34]. Nevertheless, considering the titers of antimicrobial compounds typical of cinnamon found in C-PEW (Table 2), their amount may be considered neglectable or anyhow bearable by different fungal cell factories. 

We focused our attention on the fungal species enlisted in Figure 3 (further details in the Materials and Methods section): all of them are classified as yeasts except for *Aureobasidium pullulans*, that is considered a yeast-like fungus [35]. The choice to test fungal clades that involve yeasts is related to the fact that those microorganisms are reliable and robust cell factories, able to convert residual biomasses into valuable compounds or to be used as chasses for further genetic modification, expanding the portfolio of putative products [36,37]. Although bacteria could be considered relevant microbial cell factories too, the higher general tolerance of yeasts towards low pH, likewise the biomass of interest, led them to be excluded in this study. As shown in Figure 3, all the tested strains were able to grow on agar plates containing C-PEW at pH 3.5, with a qualitative slower growth by *Komagataella phaffii* compared to the other tested species. These observations clearly highlight that C-PEW can be used as feedstock by a variety of fungal cell factories that possess peculiar characteristics of industrial interest. *A. pullulans* naturally produces antimicrobial compounds, industrial enzymes, and polymers such as pullulans and poly(β-L-malic acid) [35]. *K. marxianus, K. pastoris*, and *S. cerevisiae* are known to be chasses for recombinant molecules with several available tools for this purpose [38,39,40], therefore expanding the potential uses of C-PEW. *Z. bailii* and *Z. parabailii* were selected due to their ability to grow at low pH and to withstand weak organic acid: the availability of technologies for these species is increasing their applicability [41,42]. *S. stipitis* is known for producing shikimate, which can act as building blocks for high-value aromatics [43]. *C. curvatus* and *Rhodotorula glutinis* (together with *R. toruloides*) are oleaginous yeasts able to accumulate from 20% *w/w* lipids of their dry cell mass [44,45]. Therefore, the obtained single cell oils (SCOs) can be exploited for several applications, from biodiesel to waxes, as an alternative to both petrochemical and vegetable oil industries [46,47]. In addition, *R. glutinis* and *R. toruloides* are natural producers of carotenoids [17,48], a group of molecules widely used in the feed, food, dietary supplements, and dye industries with high market demand [49] that perfectly matches the bioeconomic logic of cascading [50]. Both these carotenogenic yeasts were able to produce carotenoids when fed with cinnamon waste material hydrolysate since their pinkish color was clearly visible (Figure 3). Due to the positive test of all these species, we can infer that they would be able to reproduce the same features in batch mode as well, providing a great added value to the residual biomass obtained from cinnamon processing.

Taken together, the data obtained showed that a broad range of fungal species are phenotypically able to grow on the hydrolysate of cinnamon waste material and that therefore they can be considered a starting material for bioprocesses to obtain molecules of industrial relevance, from carotenoids to recombinant proteins. 

## 4. Conclusions

The agri-food sector is leading to the accumulation of relevant amounts of residual biomasses, which are mainly burned for energy, used as animal feed, or disposed of as wastes. Nevertheless, these biomasses often contain energy-rich molecules that can still be exploited to obtain high-value products. Furthermore, the valorization of these biomasses fosters the transition from linear economies to circular ones, where the life cycle of natural resources is extended thanks to their further processing to extract additional values. In this work, we focused our attention on the waste material of the cinnamon supply chain, obtained from the polyphenol extraction (called C-PEW), which is not used for any purpose at the moment. Here, for the first time, a cinnamon-derived biomass was hydrolyzed using a commercial enzymatic cocktail, with a 33% yield of sugars on the total available, demonstrating its potentiality as feedstock for a bioprocess based on microbial cell factories.

We focused our attention on the yeast *Rhodosporidium toruloides*, naturally able to produce carotenoids and high-value molecules used in the food, the feed, the dye, and the cosmetic sectors. Since most of the carotenoid market is satisfied by chemical synthesis, there is an increasing demand for molecules of renewable origin. Here, in a separated hydrolysis and fermentation (SHF) process, *R. toruloides* was able to consume sugars from C-PEW and to accumulate up to 2.0 ± 0.23 mg/L of carotenoids. Although the obtained titer was comparable with those of other processes based on residual biomasses valorized by *R. toruloides*, the low sugar consumption in these conditions was a limiting element. An additional nitrogen source, preferentially of residual origin, might be considered to overcome this shortcoming.

In this work, it is shown for the first time the possibility to develop bioprocesses based on cinnamon-derived biomasses to provide additional production streams, such as carotenoids, to be applied as additive in sectors in which industries dealing with this residual biomass are already involved, thus increasing the value thereof. The successful test of several microorganisms of industrial relevance for their ability to grow on C-PEW hydrolysate (withstanding its low pH and anti-microbial compounds) widens the possible exploitations of this residual biomass. In addition, several new synthetic biology tools are becoming available for non-Saccharomyces fungal species, thus expanding the ability of these microbes to be tailored for industrial processes. In this case, a desirable implementation should be the engineering of *R. toruloides* to withstand antifungal molecules and to produce fine chemicals of industrial relevance. This in turn would increase the appeal of cinnamon derived biomasses, although it is a currently underrated feedstock for microbial cell factories.

Overall, this work can pave the way for further development of bioprocesses based on the exploitation of a side-stream biomass derived from the cinnamon industry, that usually would be disposed of as waste material, in order to produce not only high-value added products such as carotenoids in *R. toruloides* but also compounds from other yeast cell factories in the scenario of the cascading principles of biorefineries. 

## Figures and Tables

**Figure 1 ijerph-18-01146-f001:**
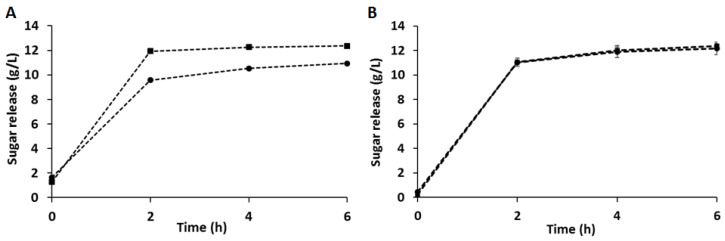
Enzymatic hydrolysis of cinnamon derived biomasses: the effect of the NS22119 cocktail (11.9% *w/wbiomass*) on 9% *w/v* cinnamon bark (CB) (**A**) and cinnamon waste material after polyphenol extraction (C-PEW) (**B**) at pH 6 (squares) or at their original pH after pretreatment (pH 4.5 for panel A, pH 3.5 for panel B) (circles). The values are the means of three independent experiments. In panel A, standard deviations are included, despite being hardly visible as numerically very small.

**Figure 2 ijerph-18-01146-f002:**
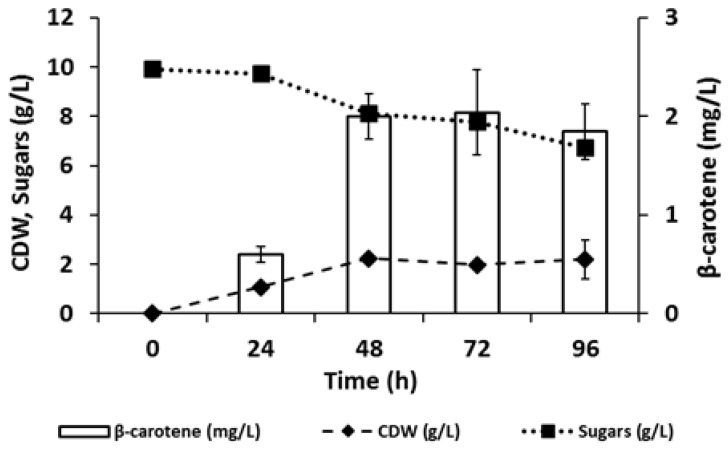
Production of carotenoids from the C-PEW hydrolysate by *R. toruloides*: cellular dry weight (CDW) in dashed line, total sugars in dotted line, and β-carotene in white bars. The values are the means of three independent experiments.

**Figure 3 ijerph-18-01146-f003:**
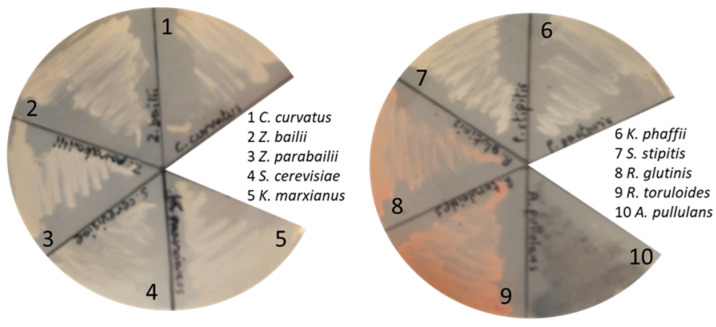
Testing of different fungal cell factories on agar plates containing only the enzymatic hydrolysate of 9% *w/v* C-PEW (pH 3.5) as a nutrient source: the picture was taken after 72 h of growth at 30 °C.

**Table 1 ijerph-18-01146-t001:** Total hydrolysis of cinnamon: cinnamon bark and waste extract hydrolysate composition following acid treatment.

Total Acid Hydrolysis of Cinnamon		
Component	Cinnamon Bark (CB)	Cinnamon Waste Material (C-PEW)
Water	19.4 ± 1.34%	/
Insoluble fraction	41.1 ± 1.26%	44.5 ± 1.86%
Acetate	1.5 ± 1.03%	2.8 ± 0.13%
Sugars	37.3 ± 0.83%	41.5 ± 1.28%
of which		
Glucose	25.2 ± 0.53%	27.2 ± 0.99%
Fructose	9.1 ± 0.15%	10.7 ± 0.32%
Arabinose	3.0 ± 0.19%	3.6 ± 0.04%

**Table 2 ijerph-18-01146-t002:** Effect of pH on enzymatic hydrolysis: the titer of sugars released by the NS22119 cocktail (11.9% *w/wbiomass*) on 9% *w/v* CB or C-PEW at different pH after 6 h of treatment. The values are the means of three independent experiments.

Enzymatic Hydrolysis of Cinnamon-Derived Biomasses	Cinnamon Bark (CB)				Cinnamon Waste Material (C-PEW)			
Component Titer (g/L)	pH 4.5	Yield	pH 6	Yield	pH 3.5	Yield	pH 6	Yield
Sucrose	0.9 ± 0.01	-	1.06 ± 0.00		0.8 ± 0.04	-	0.9 ± 0.01	-
Glucose	7.5 ± 0.09	32.98%	8.2 ± 0.08	36.07%	8.8 ± 0.42	36.7%	9.0 ± 0.12	35.9%
Fructose	2.6 ± 0.05	30.73%	2.4 ± 0.01	28.44%	1.8 ± 0.05	18.6%	1.8 ± 0.06	18.6%
Arabinose	-	-	0.8 ± 0.01	25.65%	0.8 ± 0.02	8.4%	0.7 ± 0.06	5.6%
Total sugars	10.9 ± 0.11	32.4%	12.4 ± 0.10	36.67%	12.2 ± 0.52	33%	12.4 ± 0.23	32.5%

**Table 3 ijerph-18-01146-t003:** Aromatic compounds of cinnamon origin: evaluation of the presence of aromatic compounds typical of cinnamon in CB or C-PEW hydrolysates. The values are the means of three independent experiments.

Aromatic Compounds in Cinnamon-Derived Hydrolysates	CB Hydrolysate ZZZ(W/O Autoclave Pre-Treatment)	CB Hydrolysate	C-PEW Hydrolysate
Component	Titer (mg/L)	Titer (mg/L)	Titer (mg/L)
4-hydroxybenzoic acid	-	1.8 ± 0.52	4.8 ± 0.47
p-coumaric acid	6.7 ± 0.24	2.5 ± 1.27	0.8 ± 0.16
Cinnamic acid	35.6 ± 0.51	23.2 ± 2.88	6.7 ± 1.29
Cinnamaldehyde	155.5 ± 12.55	73.4 ± 2.82	5.5 ± 0.96

## Data Availability

Not applicable.

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
