# Peer review of "Enzymatic Hydrolysate of Cinnamon Waste Material as Feedstock for the Microbial Production of Carotenoids"

_ijerph, 2021, doi:10.3390/ijerph18031146_

Round 1
Reviewer 1 Report
Type of manuscript: Article
Title: Enzymatic hydrolysate of cinnamon waste material as feedstock for the microbial production of carotenoids
Journal: International Journal of Environmental Research and Public Health
Although the overall approach is interesting, I do not recommend the acceptance of the paper presented in given form. Paper needs revision.
In the following some comments are given which could be helpful for the improvement of the paper.
- The term ,polyphenol extraction waste material from cinnamon’ or , polyphenol waste material from cinnamon’ (C-PEW) is very confusing. It suggests that the research material is actually the material containing polyphenols. I recommend using different term, e.g. cinnamon waste material after polyphenol extraction.
- Section 1. Plant biomass and feedstock preparation contains some elements of methology that should be placed in the separate section, e.g. water percentage, chemical composition of CB and C-PEW.
- There is no description of the method for the determination of acetates, which were mentioned in Table 1.
- It was shown that the release of sugars from CB is higher (13%) when performed at pH 6.
Please explain on what basis the authors found that ‘such a limited increase would not justify the additional neutralization step’. (Line 223-225)
- Line 228, there should be: , polyphenol extraction did not impair the activity of the used enzymatic preparation’ instead of ,polyphenol extraction did not impair the enzymatic activity’.
- Line 386, there should be ,to accumulate up to 2.00 ± 0.23 mg/L of carotenoids’.
Author Response
1. The term, polyphenol extraction waste material from cinnamon’ or , polyphenol waste material from cinnamon’ (C-PEW) is very confusing. It suggests that the research material is actually the material containing polyphenols. I recommend using different term, e.g. cinnamon waste material after polyphenol extraction.
We thank the reviewer for the suggestion. We modified the name of the biomass in “cinnamon waste material after polyphenol extraction” along the text, maintaining the acronym C-PEW, whose meaning is explained in lines 69-70.
2. Section 1. Plant biomass and feedstock preparation contains some elements of methology that should be placed in the separate section, e.g. water percentage, chemical composition of CB and C-PEW.
We modified the title of the section with Plant biomass: feedstock preparation and analysis: we think that keeping together these pieces of information is important in order to underline the fact that the C-PEW we are using is a biomass deriving from a previous and specific extraction process.
3. There is no description of the method for the determination of acetates, which were mentioned in Table 1.
In the section 2.5 Analytical methods we describe that acetic acid was measured by HPLC, together with glucose, sucrose, arabinose, fructose.
4. It was shown that the release of sugars from CB is higher (13%) when performed at pH 6. Please explain on what basis the authors found that ‘such a limited increase would not justify the additional neutralization step’. (Line 223-225)
In order to increase the pH to the optimum for the enzymatic cocktail (pH 6) starting from the lower pH of the waste materials (pH 3.5) the addition of a strong base (e.g. NaOH) is needed. Therefore, this step may impair both economic and environmental sustainability of the overall process, that would not be balanced by the scarce additional amount of sugar released at pH 6.
In order to underline better this comment, we added in the text that by increasing the pH we incremented the total sugar by just 1 g/L (Lines 236-7), therefore limiting the impact of a costly additional step.
5. Line 228, there should be: , polyphenol extraction did not impair the activity of the used enzymatic preparation’ instead of ,polyphenol extraction did not impair the enzymatic activity’.
6. Line 386, there should be ,to accumulate up to 2.00 ± 0.23 mg/L of carotenoids’.
We modified both lines accordingly.
Reviewer 2 Report
In the work, the authors described a bioprocess that enzymatically hydrolyzed cinnamon waste (C-PEW) to produce carotenoids by Rhodosporidium toruloides. Interestingly, although there are some typical antimicrobial compounds in hydrolysate such as aromatic 4-hydroxybenzoic acid, p-coumaric acid, cinnamic acid and cinnamaldehyde, R. toruloides and other fungal cell factories still grew well, implying great potential in cinnamon waste biorefinery. It is a simple story with a very long chain, but makes sense. The authors’ data basically supports their point of view, but some very important data need to be supplemented, and some issues need to be explained clearly.
Some specific comments
- The authors need to reorganize the introduction to highlight the importance and innovation of the article, especially the difficulty and novelty of utilizing cinnamon waste (due to composing of typical antimicrobial compounds).
- In figure 1 and Table 2, the concentration of total sugars in hydrolysate is low, which may limit subsequent fermentation phenotype. Higher solid loading (for example, 15% or 18% w/v) is recommended for further investigation.
- In page 6-7, the authors measured the aromatic compounds component in CB and C-PEW hydrolysate. I suggest the authors supplement some data about growth inhibition of these compounds to typical bacteria and yeast ( coli and Saccharomyces cerevisiae), as well as the selected R. toruloides in the work. These data could somewhat explain why R. toruloides utilize C-PEW hydrolysate well. If the authors could measure the concentration of these compounds during fermentation in Figure 2, it would be much better. Besides, why R. toruloides stopped grow after 48 h?why sugars concentration no longer decreased after 96 h? because of the inhibitors? The authors should provide a control experiment that R. toruloides grows in pure sugars (with same sugar component), so that the readers could follow the logic and know more information.
- More details such as colonies photograph or colony number statistics should be presented for Table 4. Otherwise, I cannot make a rational or emotional judgment of Figure 4.
- The shortcomings of this work and future optimization plans need to be discussed. In addition, the promotion and challenges of some recently reported toruloides that can efficiently utilize different biomass or agricultural wastes need to be discussed. In particular, the rapid development of synthetic biology has an important impact on pretreatment-enzymatic hydrolysis-fermentation process.
Author Response
1. The authors need to reorganize the introduction to highlight the importance and innovation of the article, especially the difficulty and novelty of utilizing cinnamon waste (due to composing of typical antimicrobial compounds).
We modified lines 67-71 in order to underline the novelty of the proposed bioprocess, and the possibility of exploiting an underrated biomass as C-PEW.
2. In figure 1 and Table 2, the concentration of total sugars in hydrolysate is low, which may limit subsequent fermentation phenotype. Higher solid loading (for example, 15% or 18% w/v) is recommended for further investigation.
High solid loadings may impair the production of carotenoids, since these molecules are secondary metabolites whose accumulation is exacerbated during the stationary phase. Therefore, higher amount of sugars would delay the insurgency of the stationary phase and as a consequence the production of carotenoids, thus decreasing the productivity. In addition, titers of C-PEW higher than 9% resulted in problems during the homogenization of the hydrolysis solution, because the insoluble fraction tended to float in water. Nevertheless, we thank the reviewer for the suggestion, because it paves the way for further development of the process.
3. In page 6-7, the authors measured the aromatic compounds component in CB and C-PEW hydrolysate. I suggest the authors supplement some data about growth inhibition of these compounds to typical bacteria and yeast (E. coli and Saccharomyces cerevisiae), as well as the selected R. toruloides in the work. These data could somewhat explain why R. toruloides utilize C-PEW hydrolysate well. If the authors could measure the concentration of these compounds during fermentation in Figure 2, it would be much better. Besides, why R. toruloides stopped grow after 48 h?why sugars concentration no longer decreased after 96 h? because of the inhibitors? The authors should provide a control experiment that R. toruloides grows in pure sugars (with same sugar component), so that the readers could follow the logic and know more information.
As kindly suggested by the reviewer, we performed new kinetic of growth providing R. toruloides with a synthetic medium supplemented with the same amount of sugars present in C-PEW hydrolysate, at both pH 3.5 and 5.5 (optimal for this yeast). As shown in Figure S1, R. toruloides was able to consume completely the sugars and to reach higher CDW than in C-PEW hydrolysate, regardless from the pH of the medium. Regarding inhibitory compounds typical of cinnamon origin, after 24 hours of fermentation their presence was not detected anymore: this fact could be related to possible detoxification operated by the yeast itself. The reduced growth and sugar consumption in C-PEW hydrolysate could be instead ascribed to the low initial amount of nitrogen source, which resulted to be depleted after 48 hours (Figure S2), in correspondence to the stationary phase of growth. As inhibitory compounds typical of cinnamon origin were not detected after 24 hours, nitrogen depletion should be considered the main bottleneck for R. toruloides growth in C-PEW hydrolysate (lines 293-301). We also added information regarding minimal inhibitory concentration of cinnamaldehyde towards several microbial pathogens in section 3.4 (Lines 332-6)
4. More details such as colonies photograph or colony number statistics should be presented for Table 4. Otherwise, I cannot make a rational or emotional judgment of Figure 4.
We modified the part related to the test of various fungal species on C-PEW hydrolysate, substituting Table 4 with Figure 3, which displays directly the picture of the plates.
5. The shortcomings of this work and future optimization plans need to be discussed. In addition, the promotion and challenges of some recently reported toruloides that can efficiently utilize different biomass or agricultural wastes need to be discussed. In particular, the rapid development of synthetic biology has an important impact on pretreatment-enzymatic hydrolysis-fermentation process.
We added recent examples of carotenoids production from residual biomasses by R. toruloides in lines 302-4, highlighting the possibility to further increase the production if more glucose would be consumed. In addition, main shortcomings of the process as well as the innovative scenarios widened by synthetic biology are now discussed in the conclusion section (lines 392-96 and 402-7)
Reviewer 3 Report
The authors of the manuscript "Enzymatic hydrolysate of cinnamon waste material as 2 feedstock for the microbial production of carotenoids" report the separate enzymatic hydrolysis and fermentation of a lignocellulosic biomass, cinnamon waste material, also proposing its valorization as feedstock for the production of carotenoids or the growth of other yeasts of industrial interest.
It is a very interesting, nicely written and well-conducted study. I have the following doubts/requests:
MAJOR POINTS:
1) The study lacks a comparison with previous similar studies in terms of efficiency of the hydrolysis and growth of yeasts on hydrolyzates
2) The authors have chosen as pretreatment only an auto-hydrolysis (l.112-113). Why not an acid- or a base-catalyzed pretreatment? Usually, they are much more effective. Moreover, this could have allowed the use of a lower amount of enzyme in the hydrolysis process.
3) From what I understand, the hydrolysates have been used as growth media (l.141 and l.207). If so, why no other nutrient (e.g. Nitrogen, yeast extract) was added to obtain a higher biomass and carotenoid production? I understand that this could represent an additional cost that however could have been compensated by much higher production. Moreover N is very low (l. 233) for a growth medium and the authors suggest that the low glucose consumption might be ascribed to a lack of nitrogen (l. 288)
4) It is not clear why solid media were used to assess the growth of the yeasts reported in Table 4
MINOR POINTS:
1) l.78; when high added-value products are obtained it is not useful to point-out yields which are usually very low
2) l.151; "...comparing.." should be "by comparison..." or "by comparing them..."
3) l.157; acetonitrile is usually indicated by CH3CN
4) Table 1; I assume lignin is in the insoluble fraction; is there a way to obtain its value? Please explain why C-PEW has no water at all
5) l. 231; it is not clear if the yield is referred to CB or to C-PEW
6) legend of Figure 1; it could be better to write " Enzymatic hydrolysis of cinnamon derived biomasses. Effect of the NS22119 cocktail (11.9% 236 w/w biomass) on 9% w/v CB (A) and C-PEW (B) at pH 6 (squares) or at their original pH after pre-treatment (pH 4.5 for panel A, pH 3.5 for panel B) (circles). Values are the means of three independent experiments."
7) standard deviations should be reported in Figure 1
8) Table 2; it should be better to report also the % to better compare this data to that of Table 1 (of course considering the hydrolysis of sucrose)
9) Table 3; it is not clear what exactly is "CB hydrolysate (w/o) pretreatment"; is the auto-hydrolysis step missing here? in this case, are these molecules released after the enzymatic hydrolysis? Values are much higher in this case than in the case of the hydrolysates. Please clarify.
10) l. 301; please be generic without specifying the company that provided the biomass
Author Response
MAJOR POINTS:
1) The study lacks a comparison with previous similar studies in terms of efficiency of the hydrolysis and growth of yeasts on hydrolyzates
We agree with this comment by the reviewer; the reason why we did not add such comparisons is that the sole work (at the best of our knowledge) that showed enzymatic hydrolysis of cinnamon ([21], Tang et al., 2019) states that 22.2 ± 0.61 g/L of glucose were obtained from 10% of cinnamon bark enzymatically hydrolyzed. Unfortunately, the authors did not specify if that % is weight/volume or something else, and they do not express the volume of hydrolysis: therefore, it is not possible to know from how much biomass that amount of glucose was released. Consequently, a comparison with the data here shown is not feasible. In addition, there are no other examples in literature of yeast growth for the development of bioprocesses on these cinnamon-derived biomasses.
2) The authors have chosen as pretreatment only an auto-hydrolysis (l.112-113). Why not an acid- or a base-catalyzed pretreatment? Usually, they are much more effective. Moreover, this could have allowed the use of a lower amount of enzyme in the hydrolysis process.
Chemical-catalyzed pretreatments are definitely more effective rather than auto-hydrolysis in autoclave. We decided not to involve their use in this work for several reasons. The work was focused on the overall green exploitation of cinnamon, as also the phenolic extraction that occurs upstream and generates this residue is intended to minimize environmental impacts. We aimed at the reduction in the use of chemical also for the hydrolysis, therefore we tried to avoid as much as possible their involvement. For the sake of clarity we added this statement in the section 2.2. In addition, here we show that the initial acidic pH of the biomass was still compatible with the activity of the enzymatic cocktail: the use of acid pretreatment generally drops pH to lower values, which could have impaired the enzymatic catalysis, making the neutralization step mandatory. In addition, autoclaved auto-hydrolysis permitted to sterilize the biomasses as well, thus avoiding any contamination of the media, causing undesirable fermentation.
Regarding the quantity of enzymes used, the high amount was pretextual to show the maximum capacity of the hydrolysis: this paves the way for further implementation of this step with lower amount of enzymes. This statement was added in section 2.2 as well.
3) From what I understand, the hydrolysates have been used as growth media (l.141 and l.207). If so, why no other nutrient (e.g. Nitrogen, yeast extract) was added to obtain a higher biomass and carotenoid production? I understand that this could represent an additional cost that however could have been compensated by much higher production. Moreover N is very low (l. 233) for a growth medium and the authors suggest that the low glucose consumption might be ascribed to a lack of nitrogen (l. 288)
As underlined by the reviewer, nitrogen is a key element for microbial growth, therefore the provided media must contain at least a N source. The goal of the work was to provide exclusively this residual biomass, without being dependent on external additional carbon or nitrogen sources, both to show the proof of concept of the process and to provide to the industrial partner with a process that could be developed exploiting what is derived only from the main stream of the cinnamon value chain. Of course, the addition of other residual biomasses rich in nitrogen (i.e., corn steep liquor) could be considered as well, but they did not match with the typical supply chains of spices. In addition, a low amount of nitrogen compared to carbon is important to trigger the production of lipids (like carotenoids) in oleaginous yeasts like R. toruloides, therefore a high C/N ratio could be beneficial for the production of interest. Furthermore, carotenoids yields obtained from C-PEW are comparable with those from the use of another residual biomass (namely Camelina sativa meal) with R. toruloides that we previously obtained without any addition of nitrogen ([18] Bertacchi et al., 2020), witnessing the ability of this yeast to use residual biomasses as growth medium. In order to indicate nitrogen supply as a key point for further development of the process, we added a statement in the Conclusions (lines 392-6).
4) It is not clear why solid media were used to assess the growth of the yeasts reported in Table 4
Solid media in agar plates was preferred to liquid media in shake flasks in order to quickly visualize the ability of several fungal species to grow on C-PEW hydrolysate, as a proof of concept of the adaptability of this biomass. This test paves the way for further and more accurate experiments on the use of the most promising microbial cell factory, to be investigated in liquid media. We added figure 3 instead of Table 4 in order to provide an immediate impression of the experiment and avoid a dubious description, as the reviewer underlined
MINOR POINTS:
1) l.78; when high added-value products are obtained it is not useful to point-out yields which are usually very low
We understand the fact that the values of such products do not impose high yields, nevertheless, we think that yields are important to describe the process itself, and to understand how to further improve it. We added a statement in the text to clarify our point (lines 305-6).
2) l.151; "...comparing.." should be "by comparison..." or "by comparing them..."
3) l.157; acetonitrile is usually indicated by CH3CN
We modified accordingly both lines in the new text.
4) Table 1; I assume lignin is in the insoluble fraction; is there a way to obtain its value? Please explain why C-PEW has no water at all
As the reviewer correctly stated, lignin is part of the insoluble fraction. We did not further investigate its value because we were interested in the fermentable components, such as sugars. C-PEW has no water because this biomass is dried after the extraction of polyphenols. To make it clearer we modified line 97 accordingly.
5) l. 231; it is not clear if the yield is referred to CB or to C-PEW
We added “from C-PEW” in line 243.
6) legend of Figure 1; it could be better to write " Enzymatic hydrolysis of cinnamon derived biomasses. Effect of the NS22119 cocktail (11.9% 236 w/w biomass) on 9% w/v CB (A) and C-PEW (B) at pH 6 (squares) or at their original pH after pre-treatment (pH 4.5 for panel A, pH 3.5 for panel B) (circles). Values are the means of three independent experiments."
We thank the reviewer for this suggestion, we modified the text accordingly.
7) standard deviations should be reported in Figure 1
Standard deviations are present in the figure, but they have little value, therefore it is difficult to spot them. We specified directly in the legend of Figure 1.
8) Table 2; it should be better to report also the % to better compare this data to that of Table 1 (of course considering the hydrolysis of sucrose)
We included in Table 2 this information as well.
9) Table 3; it is not clear what exactly is "CB hydrolysate (w/o) pretreatment"; is the auto-hydrolysis step missing here? in this case, are these molecules released after the enzymatic hydrolysis? Values are much higher in this case than in the case of the hydrolysates. Please clarify.
The statement of the reviewer is correct: in this case we did not autoclave the biomass, to underline that this step eliminates part of the possible inhibitory compounds of cinnamon origin. For the sake of clarity, we added “autoclave” in the table.
10) l. 301; please be generic without specifying the company that provided the biomass
We substituted the name of the company with “Industries in this sector”.
Round 2
Reviewer 2 Report
The author has responded to my question carefully, and I would like to recommend to accept it.